# SARS-CoV-2 genomic surveillance in wastewater as a model for monitoring evolution of endemic viruses

Mukhlid Yousif [1,2] ✉, Said Rachida[1], Setshaba Taukobong[1], Nkosenhle Ndlovu[1], Chinwe Iwu-Jaja[1], Wayne Howard [1], Shelina Moonsamy [1], Nompilo Mhlambi[1], Sipho Gwala[1], Joshua I. Levy [3], Kristian G. Andersen [3], Cathrine Scheepers[4], Anne von Gottberg [5,6], Nicole Wolter [5,6], Jinal N. Bhiman [5,6], Daniel Gyamfi Amoako[5], Arshad Ismail[7,8], Melinda Suchard [9] & Kerrigan McCarthy[1,2]

As global SARS-CoV-2 burden and testing frequency have decreased, wastewater surveillance has emerged as a key tool to support clinical surveillance efforts. The aims of this study were to identify and characterize SARS-CoV-2 variants in wastewater samples collected from urban centers across South Africa. Here we show that wastewater sequencing analyses are temporally concordant with clinical genomic surveillance and reveal the presence of multiple lineages not detected by clinical surveillance. We show that wastewater genomics can support SARS-CoV-2 epidemiological investigations by reliably recovering the prevalence of local circulating variants, even when clinical samples are not available. Further, we find that analysis of mutations observed in wastewater can provide a signal of upcoming lineage transitions. Our study demonstrates the utility of wastewater genomics to monitor evolution and spread of endemic viruses.

Since SARS-CoV-2 is shed into stool[1–3] and urine[1] and is detectable in wastewater[4], quantification and sequencing of SARS-CoV-2 in wastewater has the potential to overcome many inherent limitations in clinically-based epidemiological approaches. Throughout the COVID-19 pandemic, clinical surveillance has relied on testing and sequencing of samples from infected individuals. However, when clinical testing forms the basis for surveillance, population health-seeking behavior, test accessibility and testing practices of attending clinicians limit the

generalizability of data[5]. In particular, clinical testing generally only detects symptomatic cases and testing practices often vary by location and over time[6], leading to an incomplete representation of virus spread and diversity. Wastewater-based testing for SARS-CoV-2 overcomes these limitations by sampling wastewater from the entire community, which adds key information to our understanding of SARS-CoV-2 transmission dynamics, and at a fraction of the cost of clinical surveillance. As such, over 70 countries now provide

[1]Centre for Vaccines and Immunology, National Institute for Communicable Diseases, a division of the National Health Laboratory Service, Johannesburg, South Africa. [2]Department of Virology, School of Pathology, Faculty of Health Sciences, University of the Witwatersrand, Johannesburg, South Africa. [3]Department of Immunology and Microbiology, The Scripps Research Institute, La Jolla, CA 92037, USA. [4]SAMRC Antibody Immunity Research Unit, Faculty of Health Sciences, University of the Witwatersrand, Johannesburg, South Africa. [5]Centre for Respiratory Diseases and Meningitis, National Institute for Communicable Diseases, a division of the National Health Laboratory Service, Johannesburg, South Africa. [6]School of Pathology, Faculty of Health Sciences, University of the Witwatersrand, Johannesburg, South Africa. [7]Sequencing Core Facility, National Institute for Communicable Diseases, a division of the National Health Laboratory Service, Johannesburg, South Africa. [8]Department of Biochemistry and Microbiology, Faculty of Science, Engineering and Agriculture, University of Venda, Thohoyandou, South Africa. [9]Department of Chemical Pathology, School of Pathology, University of the Witwatersrand, Johannesburg, South Africa. ✉e-mail: Mukhlid.yousif@gmail.com

monitoring and public reporting of geographical and temporal trends in wastewater SARS-CoV-2 levels[7,8].

Wastewater genomic surveillance enables monitoring of the specific variants circulating in a community. Whole genome sequencing[9] and other methods such as real-time PCR[10] enable detection and characterization of SARS-CoV-2 variants, and can be applied in wastewater samples. Recent work has shown the potential for recovery of complete virus genomes from wastewater[11], demonstrated comparable lineage dynamics via wastewater and clinical surveillance, and identified novel mutations and lineages in wastewater before appearance in clinical samples[6,12]. To date, wastewater sequencing of SARS-CoV-2 has not been widely applied in low- or middle-income countries.

South Africa is a middle-income country with a population of over 55 million persons, most of whom live in urban centers located in five of the country's nine provinces. South Africa has over 1,000 wastewater treatment plants[13] and the majority of South Africans (84%) have access to piped sanitation (flush toilets connected to a public sewerage system or a septic tank)[14]. After wastewater testing for SARS-CoV-2 was first described in South Africa in June 2020[15], the South African Collaborative COVID-19 Environmental Surveillance System (SACCESS) arose to monitor trends in SARS-CoV-2 levels in wastewater across the country[16].

Most clinical SARS-CoV-2 testing is provided to the public through an extensive network of laboratories including the National Health Laboratory Service (NHLS) that covers over 80% of the population. South Africa identified its first case of COVID-19 on the 5th of March 2020[17], and four waves of COVID-19 occurred within the first 24 months of virus introduction into the country. Following the initial SARS-CoV-2 wave, the Beta variant[18] was discovered and was dominant from November 2020 to February 2021 (second wave). The third wave (May to September 2021) was dominated by the Delta variant[19,20] and the fourth wave (November 2021 to January 2022) was driven by the Omicron BA.1 variant[21]. The National Institute for Communicable Diseases (NICD), a division of the NHLS, provides SARS-CoV-2 epidemiological surveillance data through collation of SARS-CoV-2 PCR results from public and private laboratories. The Network for Genomic Surveillance of South Africa[22] (NGS-SA) is a collaborating group of seven sequencing hubs located at tertiary or academic laboratories across the country. The NGS-SA monitors the epidemiology of SARS-CoV-2 variants in PCR-confirmed cases in South Africa and reports weekly on findings[23]. The NGS-SA provided the first global reports of the emergence of Beta and Omicron variants of concern (VOC)[18,21].

Here, we show that wastewater can be used to effectively characterize the dynamics of SARS-CoV-2 virus lineage spread and evolution in the population. Using surveillance from sentinel wastewater treatment plants in urban metros collected from April 2021 to the end of the Omicron BA.1 wave in January 2022, we demonstrate the utility of wastewater genomic surveillance to complement clinical surveillance efforts in a middle-income setting. We identify the potential strengths and limitations of wastewater genomic surveillance for SARS-CoV-2 in the South African context.

## Results
A total of 325 wastewater samples from sites listed in Table S1 were amplified and sequenced. Of those, 229 (70.5%) samples had >1 million reads and were included in the mutational analysis and the heatmap visualization. Out of the 325, 183 (56.3%) samples had >50% sequence coverage of the whole genome (10x depth) and these were used for Freyja analysis.

### Detection of SARS-CoV-2 variants from wastewater samples
Across all wastewater collection sites, we observed clear wave-like dynamics that closely paralleled trends observed in clinical genomic surveillance (Fig. 1A–C). The Beta variant dominated early samples in April 2021 until it was displaced by the Delta variant in June 2021.

Although the Alpha variant was briefly detectable in June, Delta continued to be the prevailing variant until November, at which point the Omicron variant was detected and almost immediately dominated the pathogen landscape. Freyja analyses also enabled more fine-grained analysis of individual lineages (Fig. 1B). As in clinical surveillance (Fig. 1C), AY.45 was identified as the dominant lineage during the Delta wave across South Africa, and we regularly detected low levels of C.1.2 and other B.1.1.X lineages leading up to and during the Omicron wave. We also identified cryptic circulation of A lineage viruses including A.25 as well as the Alpha-Delta recombinant lineage XC in June, neither of which had been previously reported in South Africa. During the Omicron wave, we observed the rapid rise of the Omicron BA.1 lineage, which was eventually displaced by its sister-lineage BA.2 by the end of January 2022. We also identified substantial prevalence of BA.3 and other Omicron BA.1-BA.2 recombinant lineages, including XE, XAD, and XAP, that were rarely observed in clinical surveillance during the study period.

### Comparison of wastewater and clinical genomic surveillance in Gauteng, KwaZulu-Natal, and Free State provinces
In Gauteng, Freyja analysis of wastewater identified the Beta variant as the dominant variant in April and May 2021, and in low frequency in June 2021 (Fig. 2A), closely matching clinical surveillance during the same period (Fig. 2B). The Alpha variant was detected in a small proportion of clinical samples from April to July 2021, while Freyja reported the Alpha variant in June and August 2021. In May 2021, the Delta variant was first detected both in wastewater and clinical samples, and remained dominant until October 2021 in both wastewater and clinical samples. We detected a small population of the XC recombinant in June 2021, but this was not detected in wastewater or clinical surveillance from any other provinces. Freyja reported the presence of the C.1.2 virus lineage from June 2021 until August 2021, although no wastewater samples met the sequencing quality threshold in September 2021, whereas clinical surveillance detected C.1.2 through October. In October 2021, Omicron (BA.1) was first detected in clinical specimens and rapidly became the dominant lineage. BA.2 was first detected in late November and supplanted BA.1 by January 2022. However, Omicron was dominant in wastewater from November 2021 on, with BA.1, and BA.3 lineages being detected. However, the BA.2 variant was only detected in wastewater in January 2022. BA.3 variants were reported in wastewater in November 2021 and again in January 2022 by Freyja, but in clinical isolates, BA.3 was only detected in three samples, all collected in December 2021. We also detected the BA.1-BA.2 Omicron recombinant lineages XE, XAD, and XAP.

In wastewater samples from KZN Province (Fig. 2C), the Beta variant was detected in June at low read frequencies. However, wastewater samples failed to amplify in April 2021. The Beta variant was not detected in wastewater samples in May. Amongst clinical specimens collected in KZN Province (Fig. 2C), the Beta variant was detected from April to June 2021, and was dominant until May 2021. In wastewater samples, the Delta variant dominated from May 2021 until August 2021. C.1.2 appeared in wastewater samples in June and in August. By comparison, C.1.2 appeared for the first time in clinical samples from KZN in July 2021 until November 2021 at small proportions (<10%). In May 2021, the Delta variant appeared in clinical specimens at low read frequencies in April and May 2021, and was the dominant variant from June until October 2021. The Omicron variant appeared and quickly dominated clinical samples in November 2021, although Delta variant was still detectable along with C.1.2. In December 2021 and January 2022, Omicron completely replaced all circulating lineages. In December, BA.2 was first detected in low proportions and became co-dominant with Omicron BA.1 in January 2022 in both wastewater and clinical samples. Amongst wastewater samples, Omicron was present from November 2021, although wastewater samples failed to amplify in September and October 2021. Omicron

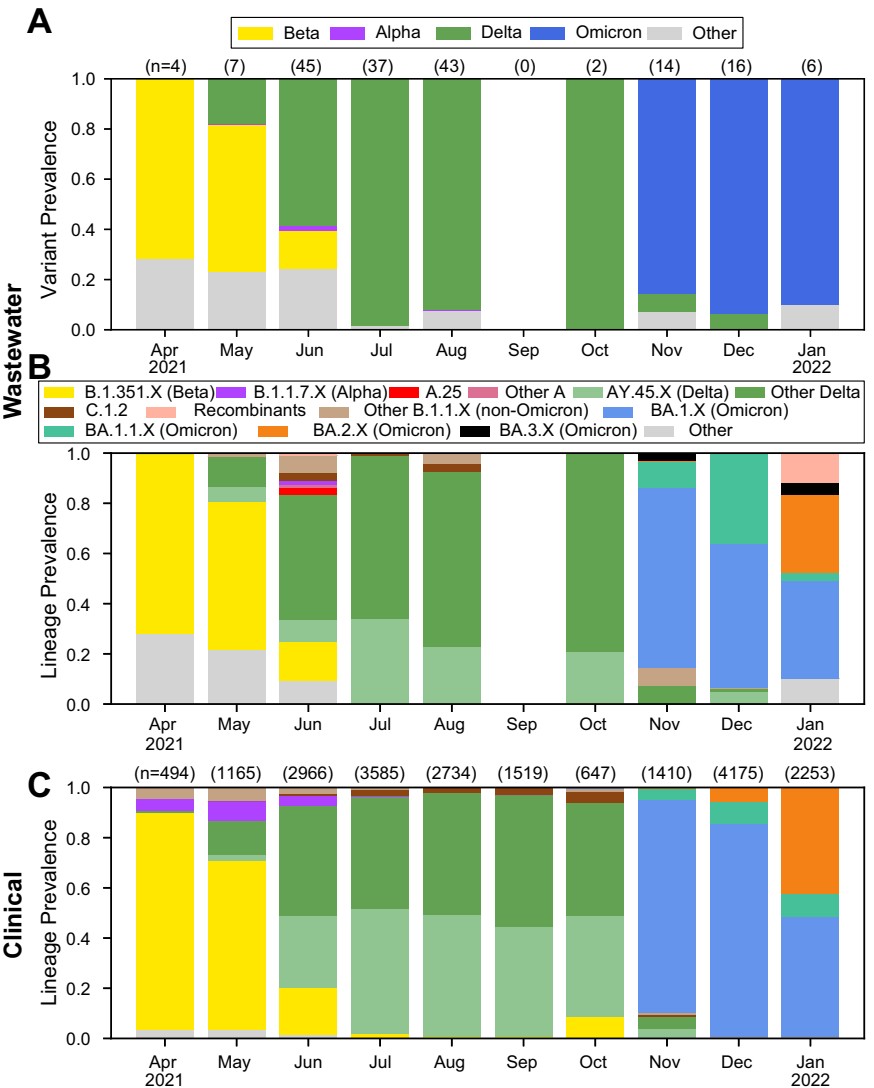

**Fig. 1 | Nationwide wastewater and clinical genomic surveillance in South Africa.** The prevalence of VOCs (**A**) and lineages (**B**) by month from wastewater samples, from April 2021 to January 2022 estimated using the Freyja tool. Only samples with sequence coverage of >50% were included. **C** Nationwide clinical genomic surveillance trends over the same period.

BA.2 was detected in wastewater samples in January 2022 for the first time, along with Omicron recombinants XE and XAP, which were also observed in Gauteng wastewater.

In Free State, wastewater, and clinical samples, the Beta variant was present from April 2021 until June 2021 but was only dominant in April and May 2021 (Fig. 2E, F). In July 2021, a small proportion of clinical samples (<5%) yielded Beta variant but wastewater samples did not show evidence of the Beta variant. The Alpha variant was present in a small proportion of clinical specimens in April, May, and June 2021, but was not detectable by Freyja. In June 2021, the Delta variant appeared in both wastewater and clinical samples and appeared co-dominant with Beta variant by both Freyja and clinical genomic analysis. C.1.2 was detected in clinical specimens from July until October 2021, but was not detected by Freyja during the same period. The Omicron variant appeared for the first time in November 2021 in clinical samples and dominated until the end of the study period. BA.2 was detected in December 2021 and January 2022. Amongst wastewater samples, Omicron appeared in November 2021, but BA.2 was not detected in wastewater samples.

Across the provinces included in study, we observed some key differences among circulating lineages observed via wastewater. The

Alpha variant was detected in Gauteng and Free State in June, but not in KZN. We also observed a small amount of Alpha (about 1% prevalence) in Gauteng in August, but not in KZN or Free State. Delta appeared dominant in the earliest samples taken from KZN, whereas Beta was clearly dominant in Gauteng and Free State. BA.1.1 appeared to play a larger role in Omicron BA.1 spread in Free State than in other provinces, and no BA.2 was observed in Free State during the study. Omicron BA.3 was only detected in Gauteng province, although other BA.1-BA.2 Omicron recombinants were detected both in Gauteng and KZN. We did not detect any BA.2 in Free State, and observed relatively little in KZN, indicative of the BA.2 wave starting earlier in Gauteng. Finally, A.25 and other A lineage circulation was not observed in Gauteng, Free State, or KZN, or in nationwide clinical surveillance, but it was observed in Eastern Cape in June, despite limited wastewater sampling from the province.

### Detection of SARS-CoV-2 variants from wastewater samples using signature mutation analysis

In addition to performing lineage assignment with Freyja, we also analyzed the frequency of variant-specific "signature" mutations, which are lineage-defining, nonsynonymous mutations in the Spike

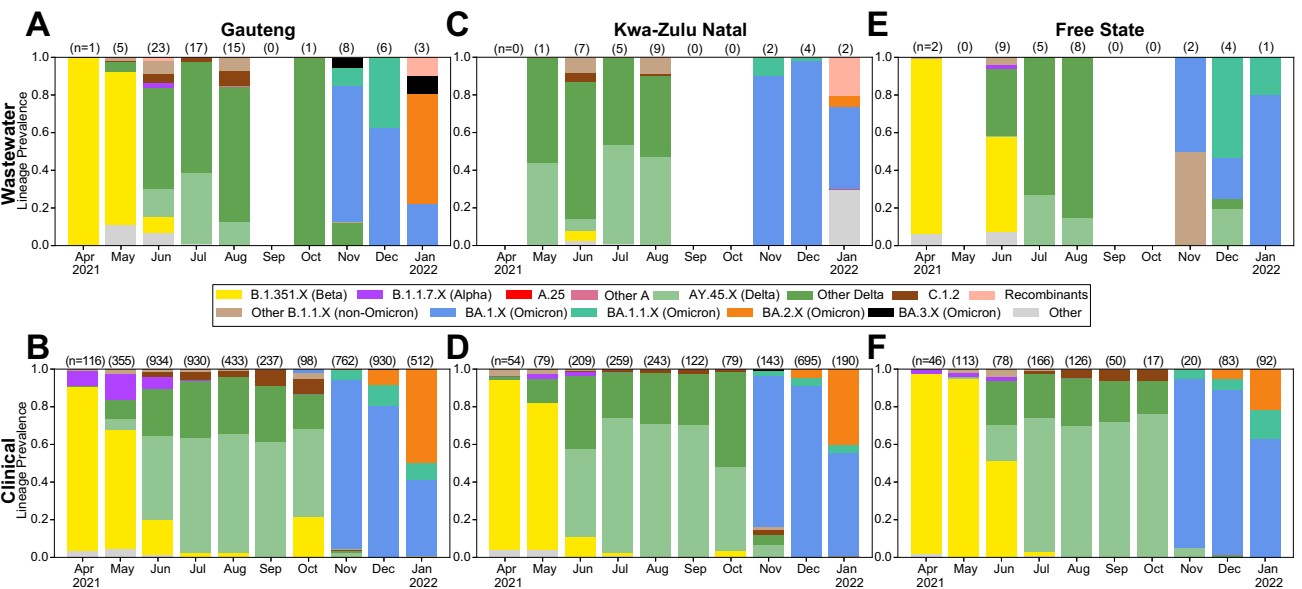

**Fig. 2 | Province-level virus population dynamics.** Wastewater lineage dynamics estimated using Freyja (top row) and clinical genomic surveillance (bottom row) from Gauteng (**A**, **B**), KwaZulu-Natal (**C**, **D**), and Free State (**E**, **F**) provinces.

region of the SARS-CoV-2 genome (Table S3). We identified signature mutations in 170 samples (52.3%), 79 samples from Gauteng, 32 from KwaZulu-Natal, 32 from Free State, 12 from Western Cape, and 15 from the Eastern Cape provinces respectively. The remaining 155 (47.7%) had no signature mutations, and thus could not be independently used to identify lineages in each catchment. We included 143 (44%) samples in our analysis from Gauteng, KwaZulu-Natal, and Free State provinces, and excluded samples from Western Cape, and Eastern Cape due to the small number of samples with signature mutations. Since no minimum genome coverage threshold was used, this approach enabled analysis of lower coverage samples not used in Freyja analyses. Using these signature mutations, we were able to identify variant waves in samples across our sites (Fig. 3A–C, supplementary figure S2). Similar to the Freyja and clinical results, the Beta variant was dominant in samples from April 2021 to June 2021 in all three provinces (Gauteng, KwaZulu-Natal, and Free State). This was followed by the domination of Delta variant and then replaced completely by Omicron BA.1 and thereafter BA.2. In Gauteng Province, Alpha was detected in May, June, and July 2021, while C.1.2 was detected from June to August 2021 (Fig. 3A). In KwaZulu–Natal province, Alpha variant mutations were detected in May, and C.1.2 was detected in August 2021 (Fig. 3B). In Free State province, we detected one mutation related to Alpha variant in June 2021. Lineage C.1.2 mutations were detected in July, and August 2021 (Fig. 3C).

**Characterization of amino acid mutations in the spike region**
A total of 411 amino acid mutations were observed in the spike protein amongst all sequenced samples. Analysis of mutation frequencies per amino acid position (Fig. 4A) demonstrated a characteristic pattern of mutations in each epidemiological wave of COVID-19. The transition from Delta variant to Omicron was characterized by a disappearance of viruses with mutations in the N-terminal domain (NTD) region (E156del, F157del, and R158G), and an appearance of viruses with mutations in the receptor binding (RBD) domain (G339D, S371L, 373, N440K, S477N, E484A, Q493R, G496S, Q498R), fusion peptide (FP) region (N764K, D796Y), and the heptad repeat 1 (HR1) region (Q954H, N969K, L9811F). Between the third and fourth wave of infection low sequence coverage of spike was observed, likely due to low caseload, and few mutations were detected. Of the 411 substitutions/ deletions detected during the study period, 68 were present at >1% prevalence.

We used the outbreak.info database to compare those mutations to known published mutations at GISAID[24] of South African sequences during the study period. Out of the 68 mutations, 58 were commonly reported (Table S4). The remaining 10 mutations (S50L, H66Y, T250S, A288T, K444T, Q498H, D627H, L828F, T859N, AND Q1201K), were detected in wastewater despite being present in <1.0% in the sequences of clinical specimens worldwide from GISAID. Further, 7 of the 10 mutations were detected in <1.0% of sequences in GISAID of South African origin (Fig. 4B), and three mutations (H66Y, T250S, and T859N) were present in a prevalence ranged between 1 and 5.5% in clinical sequences from South Africa (Table 1).

## Discussion
In this study, we demonstrated that our sequencing methodology and bioinformatics pipeline facilitated detection and genomic characterization of SARS-CoV-2 lineages present in South African wastewater treatment plants during the period April 2021 to January 2022. Our results showed similar VOC spreading dynamics to those observed amongst clinical samples, and identified lineages specific to South African outbreaks, such as the AY.45 lineage that circulated widely during the Delta wave. By analyzing the frequency of signature mutations that uniquely correspond to key variants we were able to recover lineage prevalence measurements, even from low coverage samples. We also performed unbiased analysis of amino acid mutations in the spike gene, which showed both the appearance of mutations associated with well-described lineage transitions as well as a host of amino acid mutations that were uncommon or rarely reported in clinical samples. Collectively, these findings illustrate how sequence analysis of SARS-CoV-2 in wastewater complements epidemiological findings based on clinical sequencing.

Wastewater is a complex matrix containing highly fragmented virus genomic material. Our methodology generated a high number of reads with good quality and depth when compared to other studies that have sequenced SARS-CoV-2 from wastewater[25]. Because wastewater contains a mix of RNA fragments from viral particles originating from many infected individuals, the generation of consensus sequences is generally not appropriate because the consensus sequences cannot be interpreted as representing a single viral haplotype present in individuals in population, in contrast with standard protocols for clinical sequence data. Our approach allowed for identification of

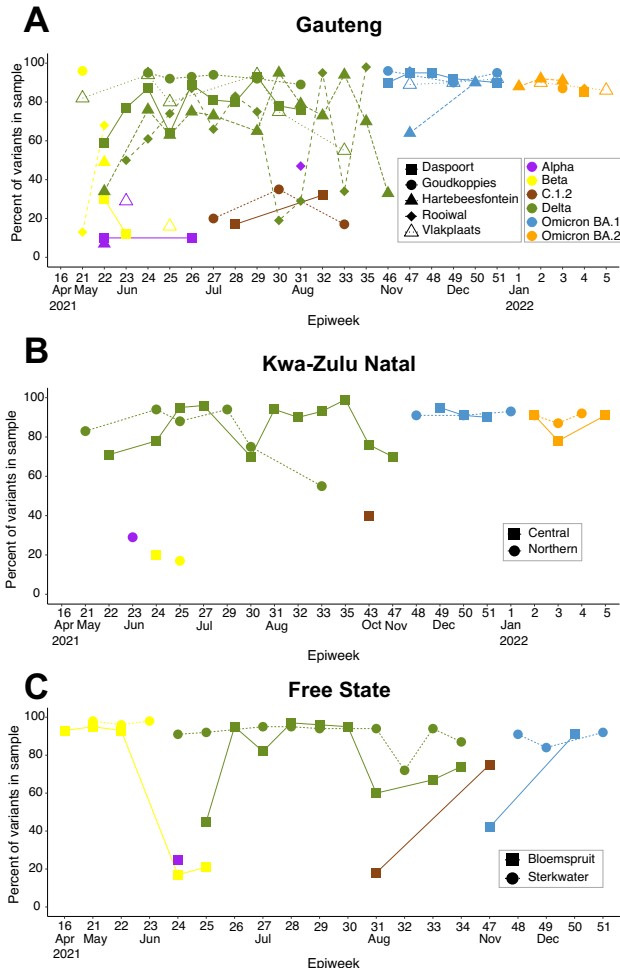

**Fig. 3 | Signature mutation analysis recovers variant waves.** Frequency of signature mutations associated with each variant per epidemiological week, for Gauteng (**A**), KwaZulu-Natal (**B**), and Free State (**C**) provinces. Signature mutations corresponding to each variant are described in Table S3.

previously described variants in wastewater samples and for detection of new patterns of mutations suggesting previously undescribed variants. We leveraged information on mutation frequency in wastewater samples (using Freyja and our signature mutation analysis) that successfully identified lineages in wastewater that corresponded to lineages identified in clinical specimens[23]. Results from Freyja were comparable to the prevalence of VOCs of SARS-CoV-2 reported from clinical specimens and additionally indicated the presence of lineages in our wastewater samples that were absent amongst sequences from clinical cases. This may be due in part to sampling biases inherent in clinical surveillance, in which only symptomatic patients are tested and of whom only a fraction was sequenced.

The Beta variant (first described in clinical specimens in South Africa in December 2020[18]) was consistently observed from the start of wastewater surveillance until the end of the third wave in epidemiological week 19 (May 2021). Similarly, the Delta variant was first seen in clinical specimens during epidemiological week 21 of 2021[20], and was first detected in wastewater samples the same week. Lineage C.1.2, described first in South Africa[26] was successfully detected in sequences from wastewater samples during week 22 to 45 in 2021 whilst clinical detections of this lineage appeared from weeks 16 to 46. In epidemiological week 46 in 2021, the Omicron variant was identified in clinical samples whilst sequences from wastewater samples also identified mutations specific to Omicron in the same week.

Our spike gene-wide mutation analysis illustrated how each variant had a distinct mutational profile of RNA sequences of the spike gene, and that this changed in each wave. Through observation of the spike protein heatmap, samples with changing profiles could often be identified before the new variant was sequenced from clinical isolates. This was clear in the transition from the Delta to the Omicron variant, where a constellation of mutations in the NTD decreased in frequency and new mutations appeared in the RBD, FP, and HR1 regions (Fig. 4A). The shifting mutational profile correlated with the increased transmissibility of the Omicron variant that led to the fourth wave of infection in South Arrfica[21].

Our mutational analysis identified multiple instances of rare mutations in the population (Fig. 4B). These mutations were found at a prevalence of >1.0% in wastewater samples but were detected at <0.001% in clinical cases based on the data from GISAID. Although they were uncommon, some mutations were reported previously. Mutation S50L was found to be associated with reduced protein stability[27]. Mutation Q498H has reportedly caused increased binding affinity of RBD to ACE2[28]. The presence of uncommon mutations could be explained as "cryptic lineages" as described by Smyth and colleagues[29], which are thought to be attributed the cryptic lineages to either unsampled, possibly chronic, infections or spillover of SARS-CoV-2 from an unidentified animal reservoir.

Sequencing of SARS-Cov-2 in wastewater currently has a number of limitations. Refining methodological approaches is essential to prevent inhibition from substances within the wastewater matrix. Where the incidence of SARS-CoV-2 is very low, virus concentration in wastewater may proceed below the level of detection, which makes it difficult to amplify and sequence the viral genome. Improvements to virus sampling and enrichment methods may offer better sequencing results, but best practices are yet to be defined, especially for low and middle-income countries. Further, emergence of new variants, such as Omicron sub-variants BA.1 and BA.2, can lead to poor primer binding and lower coverage rates, particularly in the spike protein[30], however, this itself may be considered an early warning flag of concerning viral evolution. Both scenarios render sequencing of SARS-CoV-2 from wastewater challenging. In addition, bioinformatics methods for wastewater, including mutational analysis and the Freyja tool, are currently limited by their reliance on lineage assignment based on prior clinical sequencing and publicly available sequences. On the contrary, our unbiased mutational analysis approach provides a potential tool to identify new mutations, which can used as an early warning system for the emergence of new variants.

Sequencing of SARS-CoV-2 from wastewater largely corresponded with sequencing from clinical specimens. The prevalence of VOCs and lineages in clinical specimens was shown to be detectable in wastewater during the same times, which enabled us to provide comprehensive details on VOCs and lineages in the population. Despite inherent limitations of SARS-CoV-2 sampling in wastewater, we have generated a database spanning three SARS-CoV-2 waves, that document variant and lineage changes with time and geographical location and which correspond to clinically identified variants. We have illustrated how sequences not found in clinical specimens may be identified in populations through wastewater. Our mutation-level analyses have the potential to detect new variants prior to detection in clinical samples, which may be particularly useful during times of low disease incidence between waves, when few numbers of positive clinical samples are collected and submitted for testing.

## Methods
### Wastewater sites
A total of 325 samples were collected from 15 wastewater treatment plants (WWTP) in metropolitan areas also being used for ongoing SARS-CoV-2 quantification by the NICD[31]. Sites were situated in Gauteng, Eastern Cape, Western Cape, Free State, and KwaZulu- Natal

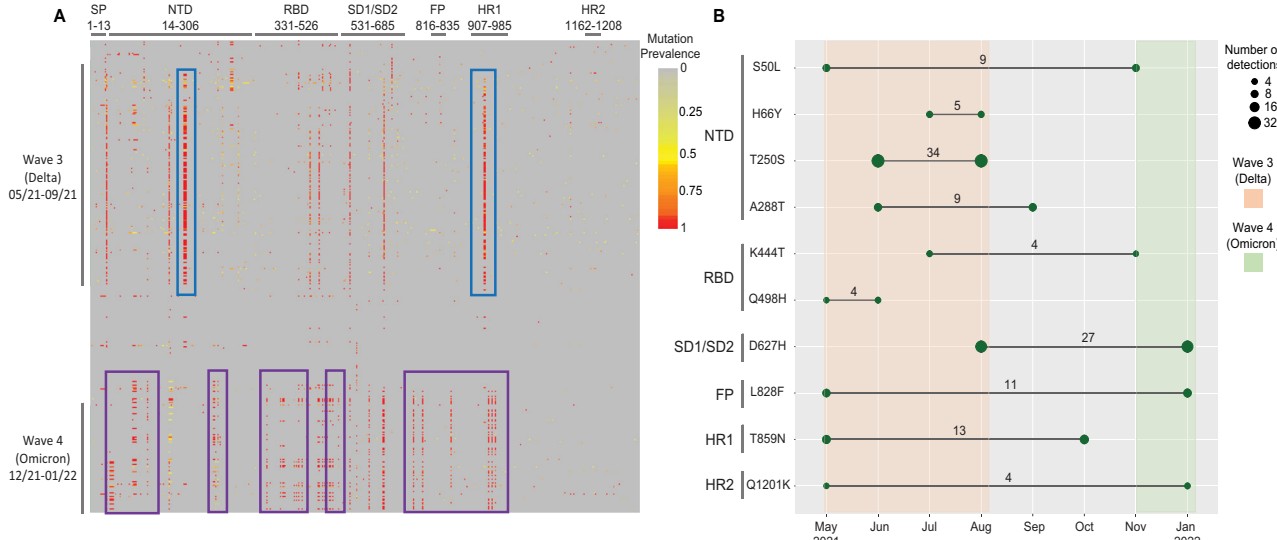

**Fig. 4 | Analyses of amino acid mutations observed in wastewater. A** Heatmap of amino acid mutations distributed across the SARS-CoV-2 spike protein in comparison with the Wuhan reference strain, arranged vertically in chronological order. Each row represents a sample, organized by the date of sample collection from earliest (top) to most recent (bottom). Each column represents an amino acid position of the spike protein annotated by region. The blue rectangle represent mutations were dominate during the third wave at the NTD and HR1, which disappeared during the fourth wave, and the purple rectangle represent new mutations introduced during the fourth wave at the SP, NTD, RBD, subdomains (SD1/ SD2), FP, and HR1. **B** Dot plot showing the uncommon mutations of SARS-CoV-2 detected in wastewater during the period April 2021 – January 2022 and their prevalence. Each row represents specific uncommon mutations, located in the named portion of the spike protein. The *x*-axis represents the periods (months) and the lines bounded by shaded circles indicate the times at which the mutations were first and last observed (as represented by a continuous line between the dates). The size of the shaded circles in each row represents the number of times the mutation was observed in the collected samples.

provinces (Figure S1). Supplementary Table S1 shows population sizes in the catchment area for each WWTP, and number of samples collected and sequenced. The samples were collected between April 2021 to January 2022 during the third and fourth waves of SARS-CoV-2 infections in South Africa.

**Sample collection, RNA extraction, amplification, and sequencing**

One liter of grab sewage samples were collected and transported to NICD at 4 °C. Viruses were concentrated from the sample by taking 200 ml and spin on a centrifuge, then 70 ml was put through a Centricon (Merck, Germany) filter for ultrafiltration[32], and RNA was extracted using the QIAamp Viral RNA kit (Qiagen, GmbH, Germany). SARS-CoV-2 was detected by RT-PCR using the Allplex™ 2019- nCoV Assay from Seegene (Seoul, Korea). RNA was re-extracted from SARS-CoV-2 positive concentrates and subjected to amplicon-based whole genome sequencing using the Sinai protocol with some modifications as described previously[33]. Paired-end libraries were prepared using Illumina COVIDSeq Kit as previously described[34] followed by sequencing (2 × 150 bp) on NextSeq 1000/2000 platform (Illumina Inc, USA).

**Sequence analysis**

**Quality control checks.** FASTQ files were trimmed, filtered based on sequence quality, assembled, and mapped to the reference genome (NC_045512.2) according to published criteria[35] using Exatype web-based bioinformatics tool (https://sars-cov-2.exatype.com/). Samples with a minimum of 1,000,000 reads, a quality Phred score of 30 or more, a sliding window of 4 and a length of 100 bp were processed for mutational analysis using ARTIC protocol (https://artic.network/ncov-2019/ncov2019-bioinformatics-sop.html) in Galaxy (https://usegalaxy.eu/)[36]. At least 10 reads were required at each nucleotide position for downstream analysis. Amino acid mutations present at 5% of reads or less were removed from the analysis. Table S2 illustrates an example of amino acid variation analysis output.

**Analysis using the Freyja tool.** To capture the dynamics of virus evolution and spread, we used Freyja[37], a tool to estimate the relative abundance of virus lineages present in wastewater. Freyja uses a "barcode" library of lineage-defining mutations to uniquely define all known SARS-CoV-2 lineages and solves for lineage abundance using a depth-weighted, least absolute deviation regression approach. Freyja is free to use and available at https://github.com/andersen-lab/Freyja. Samples with at least 50% genome coverage were included in Freyja analysis.

**Clinical samples and sequencing of SARS-CoV-2.** The NGS-SA receives randomly selected clinical samples for sequencing every week including remnant nucleic acid extracts or remnant nasopharyngeal and oropharyngeal swab samples from routine diagnostic SARS-CoV-2 PCR testing from public and private laboratories in South Africa. The NGS-SA sequencing hubs used either the Oxford Nanopore Midnight protocol or the Illumina COVIDseq Assay as described in Tegally et al[38]. All sequences are uploaded into GISAID[38] weekly. We downloaded all sequences from GISAID of South Africa origin for that were collected during the period from April 1st, 2021-January 31st 2022 excluding low coverage samples ( > 5% ambiguous bases), and used pangolin lineage assignments. We generated graphs of reported lineage by month of sample collection date using custom Python scripts.

**Sequence analysis using amino acid mutations**

**Amino acid variation analysis.** Since SARS-CoV-2 RNA in wastewater is fragmented, and fragments originate from multiple individuals (generally infected with genetically distinct viruses), the generation of consensus sequences from wastewater samples is generally not appropriate because the consensus sequences cannot be interpreted as representing a viral haplotype present in individuals in population. Rather, we inferred the presence of variants by using signature amino acid mutations. The signature mutations were defined as mutations only found in a specific variant at a prevalence of more than 1% of the total sequences deposited in GISAID. We used the amino acid variation data file generated by the ARTIC bioinformatics pipeline and R (version

Table 1 | List of uncommon mutations in the spike gene detected in wastewater samples in >1.0% prevalence, and <1.0 prevalence in sequences from clinical cases worldwide from GISAID

| Mutation | Number of samples detected in Wastewater (N = 325) | Prevalence in clinical cases in South Africa (N = 24 384) | Prevalence in clinical cases Worldwide (N = 7 744 348) | Wastewater sites |
|---|---|---|---|---|
| S50L | 9 (2,7692%) | 4 (0,0164%) | 855 (0,0110%) | Central eThekwini (3), Northern ethekwini (2), EastBank (2), Rooiwal (1), Hartbeesfontein (1) |
| H66Y | 5 (1,5384%) | 307 (1,2590%) | 2 439 (0,0315%) | Brickfield (2), Kwanobuhle (2), Rooiwal (1) |
| T250S | 34 (10,4615%) | 1 359 (5,5733%) | 1 706 (0,0220%) | Daspoort (8), Rooiwal (6), Hartbeesfontein (4), Central eThekwini (3), Northern Johannesburg (2), Northern eThekwini (2), ZandVliet (2), Bloemspruit (2), Sterkwater (2), Vlakplaats (1) |
| A288T | 9 (2,7692%) | 0 | 26 (0,0003%) | Northern eThekwini (2), Daspoort (2), Rooiwal (1), Mdantsane (1), Zandvliet (1) |
| K444T | 4 (1,2308%) | 0 | 125 (0,0016%) | Central eThekwini (3), Northern eThekwini (1) |
| Q498H | 4 (1,2308%) | 0 | 74 (0,0010%) | Central eThekiwnin (2), Zandvliet (1), Eastbank (1) |
| D627H | 27 (8,3077%) | 1 (0,0041%) | 372 (0,0048%) | Goudkoppies (4), Bloemspruit (4),Daspoort (4), Mdantsane (3), Sterkwater (3), Vlakplaats (2), Northern eThekwini (1), Northern Johannesburg (1), Rooiwal (1), Borcheds Quarry (1), Kwanobuhle (1), Central eThekwini (1) |
| L828F | 11 (3,3846%) | 3 (0,0123%) | 171 (0,0022%) | Central eThekwini (3), Zandvliet (3), Northern eThekwini (1), Northern Johannesburg (1), Hartbeesfontein (1), Rooiwal (1), Bloemspruit (1) |
| T859N | 13 (4,0000%) | 384 (1,5748%) | 30 185 (0,3898%) | Hartbeesfontein (4), Daspoort (2), Zandvliet (1), Vlakplaats (1), Central eThekwini (1), Rooiwal (1), Goudkoppies (1), Northern eThekwini (1), Sterkwater (1) |
| Q1201K | 4 (1,2308%) | 12 (0,0492%) | 2 876 (0,0371%) | Central eThekwini (3), Zandvliet (1) |

4.2.0) software to collate spike-gene mutations in a matrix such that the columns represented the amino acid positions of the spike protein and each row recorded and recorded the frequency of mutations (the proportion of all reads at that site with that mutation) for each wastewater sample. Using this list of unique mutations for each VOC and VOI in the spike protein region (Table S3) we interrogated our matrix for the presence or absence of known signature mutations in each sample using R (version 4.2.0) software. As new variants/lineages were detected and identified in the PANGO public database (https://cov-lineages.org/), we added signature mutations to the R (version 4.2.0) code, allowing us to identify the presence of new variants both retrospectively and prospectively.

**Visualization of amino acid changes using heatmap and dot blot.** Using the amino acid variations data output file from the ARTIC bioinformatics pipeline and the generated excel file that contains all amino acid variations and their respective read frequency, a heatmap was generated and interpreted to visually identify patterns of novel mutations abundance in the spike gene.

**Identification of uncommon mutations in the spike gene.** To identify the events and times at which uncommon mutations were identified in our samples, an in-house R script (R v.4.2.0) was used to generate a mutational dot plot. We defined uncommon mutations in wastewater samples as mutations that present in the Outbreak.info (https://outbreak.info/) database at less than 1% prevalence during the study period.

### Ethics approval and consent to participant

The study did not involve any human participants. An application for ethics waiver was made to the Human Research Ethics Committee of the University of the Witwatersrand and was approved (number R14/49).

### Reporting summary

Further information on research design is available in the Nature Portfolio Reporting Summary linked to this article.

## Data availability

Data (raw sequence reads) have been deposited to NCBI with accession number (PRJNA941107), and can be found in the link: https://www.ncbi.nlm.nih.gov/bioproject/?term=PRJNA941107. The findings of this study are based on metadata associated with 20,949 sequences available on GISAID up to July 13, 2023, and accessible at https://doi.org/10.55876/gis8.230825ag". The following database have been used in the study: 1. The Global Initiative on Sharing All Influenza Data (GISAID), available at: https://gisaid.org. 2. Pango lineages: latest epidemiological lineages of SARS-CoV-2, available at: https://cov-lineages.org. 3. National Library of Medicine, National Center for Biotechnology Information (NCBI), accession number: NC_045512.2. 4. Outbreak.info: a tool to explore the COVID-19 and SARS-CoV-2 data with variant surveillance reports, data on cases and deaths, and a standardized, searchable research library, available at: https://outbreak.info/.

## Code availability

All data analysis and visualization of wastewater genomic sequencing data was performed using the following: 1. Exatype v4.2.0-dev20230731 available at (https://sars-cov-2.exatype.com). 2. Bioinformatics pipeline on Galaxy, version 0.7.17.1. The pipeline uses BWA mem (Galaxy Version 0.7.17.1), samtools (Galaxy Version 1.9), iVar (Galaxy Version 1.2.2) and LoFreq (Galaxy Version 2.1.5). 3. Freyja (v1.3.10) hosted publicly on github (https://github.com/andersen-lab/Freyja) and is available under a BSD-2-Clause License (doi:10.5281/zenodo.6954863, version 1.3.10). Freyja is accessible as a package via bioconda (https://bioconda.github.io/recipes/freyja/README.html). 4.

Custom R (version 4.2.0) and Python scripts available in the project github repository (https://github.com/setshabaTaukobong/Heatmap-Matrix) (https://doi.org/10.5281/zenodo.8252094) and (https://github.com/setshabaTaukobong/Mutational-Dotplot) (https://doi.org/10.5281/zenodo.8252080).

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

## Acknowledgements

We gratefully acknowledge all data contributors, i.e., the Authors and their Originating laboratories responsible for obtaining the specimens, and their Submitting laboratories for generating the genetic sequence and metadata and sharing via the GISAID Initiative, on which this research is based. We would like to thank the local government and wastewater treatment staff for sample collection and transport. We also thank the staff of the NICD Centre for Vaccines and Immunology and the Centre for Respiratory Disease and Meningitis. special thanks to: Josie Everatt, Boitshoko Mahlangu, Anele Mnguni, Noxolo Ntuli, Gerald Motsatsi for their assistance in setting up and troubleshooting PCR testing, and ongoing supportive collaboration. We thank the team at Hyrax Biosciences for the use of their tool exatype. We would like to acknowledge the contribution from the SACCESS network. We acknowledge the financial support from the National Institute for Communicable Diseases (NICD) of South Africa, Bill, and Melinda Gates foundation (BMGF) the Water Research Commission (WRC) of South

Africa (2020/2021-00669//3), the German Society for International Cooperation (GIZ), and the donation of reagents we received from Africa CDC. MY is recipient of a grant by BMGF (INV- 036531). K.G.A. is supported by National Institutes of Health (NIH) National Institute of Allergy and Infectious Diseases U01AI151812 (WARN-ID) and National Center for Advancing Translational Sciences UL1TR002550 (CTSA). J.I.L. is funded by NIH 5T32AI007244-38.

## Author contributions

MY: co-conceptualized study, co-performed analysis, wrote, edited, and reviewed manuscript. SR: co-conceptualized study, edited, and reviewed manuscript. ST: co-performed analysis, edited, and reviewed manuscript. NN: co-performed analysis, edited, and reviewed manuscript. JIL: co-performed analysis, edited, and reviewed manuscript. MS: co-conceptualized, edited, and reviewed manuscript. KM: co conceptualized, edited, and reviewed manuscript. CI, WH, SM, NM, SG, KGA, CS, AvG, NW, JNB, DGA, AI: edited, and reviewed manuscript.

## Competing interests

All authors declare no competing interests.
