## [Peer Review File · Nature Communications]

SARS-CoV-2 genomic surveillance in wastewater as a model for monitoring evolution of endemic virusesREVIEWER COMMENTS

Reviewer #1 (Remarks to the Author):

The manuscript by Yousif and colleagues details an analysis of a large sequencing dataset of SARS-CoV-2 in wastewater in South Africa over multiple years. It is a large and relatively high quality dataset of great public health relevance.

However, it is my opinion that the manuscript itself is not currently fit for publication in Nature Communications. It is missing key methodological details, the results section is very short (<2 pages), and the figures need to be considerably reworked. Furthermore, there is no indication that the data has been shared in any form: The "data availability" section is blank. I would recommend to the authors that they submit the raw reads for this data to the NCBI SRA or EBI ENA databases.

Specific comments:

Abstract - I would recommend to the authors that they restructure the abstract so it focuses more on the results found, rather than the methodological details.

Line 54 - A heatmap is a particular visualization, not necessarily an approach into itself... I would recommend the authors rethink how to describe the actual approach they are taking in evaluating the heatmaps here in the text.

Line 59 - While there may be true of most works, it is not true in the abstract. In theory a wastewater analysis could have the potential to identify new variants, although it would be tricky. I would recommend the authors remove this sentence in the absolute form from the text.

Line 63 - please include a citation for "in stool" and "in urine" separately

Line 119 - was the entire liter really subject to ultrafiltration?

Line 124 - 150 pb should be 'bp'

Line 130 - what were the specific cutoffs used for sample inclusion of each of these quality metrics on line 130?

Line 147 - For each VOC or Variant of Interest (VOI), we identified signature single amino acid mutations by comparing the new variant/ lineage with the Wuhan reference sequence in a public database 26 (Table 2).

What about amino acid mutations that vary within an officially designated VOC? How was this handled?

Line 151 - As new variants/lineages were detected and identified in clinical specimens, we added signature mutations to the STATA code, allowing us to identify the presence of new variants both retrospectively and prospectively

What do the authors mean by this? Are they looking at particular clinical samples? Are these publicly available? Are they from particular local areas? Please specify.

Line 154 - But visualizations do not identify anything on their own - only human interpretations of them do. The authors need to make this process make clear.

Line 164 - How much do Freyja's lineage-defining mutations differ from the signature mutations identified above? Why not just use one of the two?

Results, Quality Control: The authors can simplify this section slightly, but primarily need to state how many final genomes passed quality control to be included in the analysis.

Line 176 - "Signature mutations (Table 2) were identified in 170 samples" So what variants characterize samples with no signature mutations? This doesn't really make sense to me how a sample couldn't have some variation of signature mutations. Are these samples that did not pass quality control?

From here on, the results should specify sample percentages with respect to high quality samples, not all samples, not sure if that is happening here.

Line 184 "and were successfully assigned a SARS-CoV-2 variant using the Freyja tool (Figure 2)." What does it mean by "a variant", when Freyja reports the abundances of multiple variants in a sample?

Line 190- clinical cases from where? How was this comparison performed? What data was used?

Table 1: This is routine and could be relegated to supplemental.

Table 2: This should be supplementary data or a supplementary table. Main text tables should be limited to results.

Figure 1: This figure needs to fit on a single page.

Figure 2a: why not show each sample individually? The variation within month is not shown but should be shown.

Figure 2b: There are far too many colors in this figure to be readable.

Figure 3: This figure has good potential, but needs to differentiate between not detected and poor quality. Only samples with high quality should be shown. What variants were present in the "interwave" section if no signature mutations were present?

Figure 4: What defines an "uncommon" mutation? No precise definition is provided in the text.

Reviewer #2 (Remarks to the Author):

The authors have demonstrated wastewater surveillance as a tool in detecting known variants of SARS-CoV-2. They have successfully used the tool to track the emergence of variants through the different waves of the pandemic. It becomes especially important, when the number of clinical samples that are being sequenced is very low.

Specific comments:

Figure 3: The legend needs to describe what the purple and blue rectangles represent.

Figure 4 legend: The description of X and Y axes has been switched.

Lines 272-274 and 282-286 offer contradictory conclusions of the ability of their methods in detecting novel mutations. In lines 272-274, the authors mention the need for clinical sequences to be already present in the databases as a limitation of their methods. However, in lines 282-286, they appear to suggest that their methods can identify sequences not found in clinical specimens. This contradiction needs to be addressed with respect to the methodologies employed in calling novel mutations in wastewater samples.

Reviewer #1 (Remarks to the Author):

The manuscript by Yousif and colleagues details an analysis of a large sequencing dataset of SARS-CoV-2 in wastewater in South Africa over multiple years. It is a large and relatively high quality dataset of great public health relevance.

However, it is my opinion that the manuscript itself is not currently fit for publication in Nature Communications. It is missing key methodological details, the results section is very short (<2 pages), and the figures need to be considerably reworked. Furthermore, there is no indication that the data has been shared in any form: The "data availability" section is blank. I would recommend to the authors that they submit the raw reads for this data to the NCBI SRA or EBI ENA databases.

We thank the reviewer for these suggestions and have provided substantial revisions to the document to improve our manuscript. We have re-worked figure 2A and 2B

We have also made all raw sequencing read data available via NCBI SRA (Bioproject PRJNA941107, see <https://www.ncbi.nlm.nih.gov/bioproject/PRJNA941107>).

Specific comments:

Abstract - I would recommend to the authors that they restructure the abstract so it focuses more on the results found, rather than the methodological details.

Thank you for the comment, we have amended the abstract to reflect the suggestions.

Line 54 - A heatmap is a particular visualization, not necessarily an approach into itself... I would recommend the authors rethink how to describe the actual approach they are taking in evaluating the heatmaps here in the text.

Thank you for the comment, we have amended the sentence to read as follows:

The interpretation of the heatmap was able to identify a pattern during the changes of predominate variant in wastewater with the emergence of mutations and the loss of others

Line 59 - While there may be true of most works, it is not true in the abstract. In theory a wastewater analysis could have the potential to identify new variants, although it would be tricky. I would recommend the authors remove this sentence in the absolute form from the text.

Agreed, the sentence has been removed

Background- Line 63 - please include a citation for "in stool" and "in urine" separately

This has been done

Line 119 - was the entire liter really subject to ultrafiltration?

Thanks for raising this point. It was a mistake from our side. We took 200 ml, centrifuge first, then took 70 ml to put through centricon. This has been corrected in the manuscript

Line 124 - 150 pb should be 'bp'

Thanks, this was corrected.

Line 130 - what were the specific cutoffs used for sample inclusion of each of these quality metrics on line 130?

This has been addressed. The following sentence was added to methods:

“Samples with a minimum of 1,000,000 reads, a quality Phred score of 30 or more, a sliding window of 4 and a length of 100 bp were processed for mutational analysis using ARTIC protocol”.

Another sentence was added to the section of Freyja:

“Samples with 50% sequence coverage or more were included in Freyja analysis”.

Line 147 - For each VOC or Variant of Interest (VOI), we identified signature single amino acid mutations by comparing the new variant/ lineage with the Wuhan reference sequence in a public database 26 (Table 2).

What about amino acid mutations that vary within an officially designated VOC? How was this handled?

We used a cut-off of 1% for the mutation to be present in each VOC or VOI to call it a signature mutation in the sequences from GISAID. Our analysis codes were originally developed in STATA, but we moved them to R.4.2.0. The paragraph now reads as follows:

“Rather, we inferred the presence of variants by using amino signature acid mutations. The signature mutations were defined as mutations only found in a specific variant at a prevalence of more than 1% of the total sequences deposited in GISAID. We used the amino acid variation data file generated by the Galaxy pipeline above and R.4.2.0 software to collate spike-gene mutations in a matrix such that the columns represented the amino acid positions of the spike protein and each row recorded mutations identified from a single wastewater sample”.

Line 151 - As new variants/lineages were detected and identified in clinical specimens, we added signature mutations to the STATA code, allowing us to identify the presence of new variants both retrospectively and prospectively

What do the authors mean by this? Are they looking at particular clinical samples? Are these publicly available? Are they from particular local areas? Please specify.

We refer to the public database (GISAID) in this case. We agreed that was a bit confusing and corrected accordingly. The sentence reads now as follows:

“As new variants/lineages were detected and identified in the public database, we added signature mutations to the R.4.2.0 code, allowing us to identify the presence of new variants both retrospectively and prospectively”.

Line 154 - But visualizations do not identify anything on their own - only human interpretations of them do. The authors need to make this process make clear.

Thank you for the comment, we have amended the sentence, and now it reads as follows:

“a heatmap was generated and interpreted to identify patterns of novel mutations abundance in the spike gene, after which manual linkage to known lineages was done”.

Line 164 - How much do Freyja's lineage-defining mutations differ from the signature mutations identified above? Why not just use one of the two?

Thank you for this important question. Freyja's lineage-defining mutations contain all mutations, synonymous or nonsynonymous, across the entire SARS-CoV-2 genome for every known lineage. Since we're working with whole genome sequencing data, we can leverage mutation frequency information from across the complete genome to improve our estimates of lineage prevalence in the sample.

On the other hand, the signature mutations are exclusively nonsynonymous mutations found in the Spike region of the genome and are common to all virus lineages descending from the VOC root.

These mutations are a subset of the lineage-defining mutations and are highly indicative of the presence of a specific VOC, which we visualize using our heatmap approach.

That is, although the two are intrinsically related, the lineage-defining mutations across the complete genome are key for leveraging maximum information from our sequencing analyses, while the signature mutations are well suited to visualization and VOC detection. We have now explained this more clearly in the text.

“Signature mutations, which are lineage-defining, nonsynonymous mutations in the Spike region of the SARS-CoV-2 genome that are specific to each VOC, (Table 2) were identified in 170 sample.”

Results, Quality Control: The authors can simplify this section slightly, but primarily need to state how many final genomes passed quality control to be included in the analysis.

Thank you for the comment. We have changed the section to read as follows:

A total of 325 wastewater samples from sites listed in Table S1 were amplified and sequenced. Of those, 229 (70.5%) samples had > 1 million reads were included in the mutational analysis and the heatmap analysis. Regarding sequence coverage in 10x depth, 168 (51.7%) samples had >50% sequence coverage of the whole genome and these were used for Freyja analysis

Line 176 - "Signature mutations (Table 2) were identified in 170 samples" So what variants characterize samples with no signature mutations? This doesn't really make sense to me how a sample couldn't have some variation of signature mutations. Are these samples that did not pass quality control?

After the quality control measures, we included 229 samples for mutational analysis and heatmap. Of those, 170 samples had signature mutations. The remaining 59 samples had amino acid variations but not necessarily a signature mutation.

From here on, the results should specify sample percentages with respect to high quality samples, not all samples, not sure if that is happening here.

Yes, this is correct

Line 184 "and were successfully assigned a SARS-CoV-2 variant using the Freyja tool (Figure 2)." What does it mean by "a variant", when Freyja reports the abundances of multiple variants in a sample?

In this sentence, we mean only to say that the sample was of sufficient coverage to pass quality control standards and at least one virus variant could be identified. To make this clear, we have now modified the text:

"Out of the 325 samples sequenced, 168 (51.7%) samples had a sequence coverage of >50% at 10x depth, and relative SARS-CoV-2 lineage abundances were inferred using the Freyja tool."

Line 190- clinical cases from where? How was this comparison performed? What data was used?

We compared our sequences to the sequences obtained from clinical cases which were deposited in GISAID from South Africa in the period of our analysis (April 2021 to January 2022).

Table 1: This is routine and could be relegated to supplemental.

Thank you. Table was moved to supplementary.

Table 2: This should be supplementary data or a supplementary table. Main text tables should be limited to results.

Thank you. Table was moved to supplementary.

Figure 1: This figure needs to fit on a single page.

We acknowledged these concerns. However, this is not possible. The resolution and quality of the graphs will be compromised if we put them in a single page.

Figure 2a: why not show each sample individually?

We aimed to show the variation of variant prevalence in specific timeframe (months)

The variation within month is not shown but should be shown.

The variation in each month is shown. For example, in April 2021 there was delta variant and others, while in November 2021 it was Omicron and delta.

Figure 2b: There are far too many colors in this figure to be readable.

Thank you for this comment. We have improved this figure for better visualisation.

Figure 3: This figure has good potential but needs to differentiate between not detected and poor quality. Only samples with high quality should be shown. What variants were present in the "interwave" section if no signature mutations were present?

Thank you for the question and comment. The heatmap used all amino acids variations whether signature mutations or not. Only samples with high quality were processed (more than 1 million reads, 30 or more of Phred scores).

Figure 4: What defines an "uncommon" mutation? No precise definition is provided in the text.

The uncommon mutations are mutations that have occurred in less than 1% of the total clinical datasets deposited on GISAID during the periods of April 2021 to January 2022. This has been added to the text as:

"The uncommon mutations were detected in less than 1% of the sequences from clinical specimens deposited on GISAID during the periods of April 2021 to January 2022".

Reviewer #2 (Remarks to the Author):

The authors have demonstrated wastewater surveillance as a tool in detecting known variants of SARS-CoV-2. They have successfully used the tool to track the emergence of variants through the different waves of the pandemic. It becomes especially important, when the number of clinical samples that are being sequenced is very low.

We thank the reviewer for their kind comments and agree that wastewater is of great importance, especially now as clinical testing and sequencing has become increasingly infrequent.

Specific comments:

1. Figure 3: The legend needs to describe what the purple and blue rectangles represent.

This has been addressed.

2. Figure 4 legend: The description of X and Y axes has been switched.

This has been corrected.

3. Lines 272-274 and 282-286 offer contradictory conclusions of the ability of their methods in detecting novel mutations. In lines 272-274, the authors mention the need for clinical sequences to be already present in the databases as a limitation of their methods. However, in lines 282-286, they appear to suggest that their methods can identify sequences not found in clinical specimens. This contradiction needs to be addressed with respect to the methodologies employed in calling novel mutations in wastewater samples.

Thank you for this comment. The heatmap has ability to detect new mutations but was not meant to determine variants, while the mutational analysis and Freyja determine variants but not meant to detect new mutations. To address the point, we have added the following sentence in line 27-276

“On contrary, our heatmap visualizations provide a detailed approach to monitor mutation frequencies and identify new mutations and can used as an early warning system for the emergence of new variants.”

REVIEWER COMMENTS

Reviewer #1 (Remarks to the Author):

I thank Yousif and colleagues for submitting a revised version of their manuscript. I think their dataset is large and represents an impressive set of work. However, the quality of the manuscript, the figures, and the results undermines the quality of their science in my opinion. Their results section is still too short, and as a reader you are left with multiple questions. Their Figure 1 is extremely difficult to read and needs to be simplified, combined into a single figure, and increased resolution before publication (apologies if I am looking at the wrong set of figures here). Their Figure 2 condenses their large dataset into an oversimplification, and presents no visualization or analysis of how the different locations they sampled differ in terms of their lineages. The results and details of their comparison to patient lineages is almost entirely missing, and basic results like which lineages were observed in wastewater but not in patient sampling, are missing. Reported findings like uncommon mutations are presented without basic details or interpretation. My opinion is that the science done here is important, but the manuscript would have to be greatly expanded upon and improved for publication in this journal.

Line 42 "role" should be "the role".

Line 47: remove URL from abstract.

Line 47: "We also generated a heatmap to identify patterns of emerging mutations in the spike gene."

Line 55: "by reliably predicting the circulating variants when clinical material is not available"

Line 63: Should be "Throughout the COVID-19 pandemic"

Line 65: "population health seeking behaviour, test accessibility and testing practices of attending clinicians limit the generalisability of data"

Please provide a citation discussing these potential social biases in health seeking, either for COVID-19 or more generally, beyond asymptomatic cases.

Line 121: "using the Sinai protocol with some modifications as described"

What do the authors mean by "as described"? As described in a particular cited source, or described elsewhere in this work? Please specify.

Line 131: "Again, reads were trimmed and filtered, assembled and mapped"

Line 138 "the generation of consensus sequences from wastewater samples is not meaningful"

Meaningful is a subjective term - the authors should explain what they mean by meaningful here, which is presumably that consensus sequences cannot be interpreted as representing viral haplotypes present in individuals in the population.

Line 142 "R.4.2.0" please specify as R (version 4.2) throughout.

Line 172-174 "Signature mutations, which are lineage- defining, nonsynonymous mutations in the spike region of the

SARS-CoV-2 genome that are specific to each VOC, (Table S3) were identified in 170 samples (52.3%),"

This does not make sense to me. What mutations were in the 48% of other samples that had sufficient sequencing depth? Lineages of some sort should be identifiable in all samples along with the mutations that correspond to them.

Line 175 "Figure 1 illustrates the signature mutations for VOCs that were identified in samples

from each wastewater treatment plant"

The version of Figure 1 included in this manuscript is very hard to read and is not fit for publication. It is not included in the PDF, and each panel is included as a separate tif! I do not even know what Figure 1 looks like, or how the panels are arranged. There are panels labeled A-P(!) included as separate TIFs. The panels are very hard to read as the resolution isn't adequate. The legend is only provided in a separate TIF, and there are too many different labels in my opinion. This figure has to be cleaned up in some form for publication.

Further, the text does not point to specific panels in the figure, so it's a bit unclear where the readers should actually be looking in the figure to find the data supporting the authors' claims in this section.

Section "Detection of SARS-CoV-2 variants from wastewater samples using Freyja tool"

This section is easy to follow. Figure 2 is simple and easy to follow. However, I don't understand why the authors condense their data for this figure. Why not show each sampling location separately, so all can be tracked over time? A version of figure 2 showing changes in lineage frequencies over time for each sampling site separately would communicate the primary results effectively.

I am left wondering about basic questions of the data. Are there any differences in lineages between the sampled locations? Do the authors see introductions of lineages in some sites before others? The authors could expand on their data in many ways here.

Line 187 " The proportions of lineages were shown in Figure 2b, a total of 68 lineages were detected of which 53 lineages were also reported in clinical cases and 15 lineages were not reported in clinical cases."

Where is the data for this claim? Please cite it here. If an analysis was performed, please include details and results of this analysis.

Which lineages were not reported? How many clinical genomes are the authors pulling? This could be an interesting analysis, but the authors have provided scant details about the results or the methods.

Figure 2: Please present as one figure, not separately. Please show the data for each location separately. What does "Recombinants" mean in Figure 2b?

Figure 3: Please show dates on the Y-axis.

"Figure 4 Dot blot showing the uncommon mutations of SARS-CoV-2 detected in wastewater during

the period April, 2021 – January, 2022 and their prevalence. The uncommon mutations were

detected

in less than 1% of the sequences from clinical specimens deposited on GISAID during the periods

of

April 2021 to January, 2022."

The reporting of uncommon mutations could be interesting and important, but the authors have provided no details or discussion of them. Were these rare mutations seen in other global lineages, even if they weren't observed locally? Are they consistently seen at some sites in the authors' data more than others? The authors have an opportunity to expand on this data!

Reviewer #1 (Remarks to the Author):

I thank Yousif and colleagues for submitting a revised version of their manuscript. I think their dataset is large and represents an impressive set of work. However, the quality of the manuscript, the figures, and the results undermines the quality of their science in my opinion. Their results section is still too short, and as a reader you are left with multiple questions. Their Figure 1 is extremely difficult to read and needs to be simplified, combined into a single figure, and increased resolution before publication (apologies if I am looking at the wrong set of figures here). Their Figure 2 condenses their large dataset into an oversimplification, and presents no visualization or analysis of how the different locations they sampled differ in terms of their lineages. The results and details of their comparison to patient lineages is almost entirely missing, and basic results like which lineages were observed in wastewater but not in patient sampling, are missing. Reported findings like uncommon mutations are presented without basic details or interpretation. My opinion is that the science done here is important, but the manuscript would have to be greatly expanded upon and improved for publication in this journal.

We would like to thank the reviewer for the extensive and important comments and suggestions. We have addressed these comments and suggestions by expanding on the results section to include clinical data from across South Africa, and we now compare variants collected in clinical cases versus in wastewater samples using both the Freyja tool as well as our signature mutations analysis.

Figure 1 has been changed to show the national landscape of variants in wastewater and clinical cases including variants and lineages level. Figure 2 now provides additional granularity on regional dynamics, providing a comparison of variants at provincial level in wastewater and clinical cases. Figure 3 is now a revised form of the analysis that was previously Figure 1, but now we have organized the findings by province, combining them into a single figure with vector quality resolution. We have also addressed the point of lack of details on the uncommon mutations by improving the figure, which we have now combined into Figure 4, and provided additional detail on this in the results section.

Line 42 "role" should be "the role".

Thank you, this has been corrected.

Line 47: remove URL from abstract.

Thank you, this has been removed

Line 47: "We also generated a heatmap to identify patterns of emerging mutations in the spike gene."

We have changed the title and text to show that the heatmap was a tool to visualize the pattern of changes in mutations.

Line 55: "by reliably predicting the circulating variants when clinical material is not available"

We have modified this language to make it clear that we are not predicting the variants, but rather are monitoring variant prevalence, an approach that is effective even if clinical surveillance is limited:

“We show that wastewater genomics can support SARS-CoV-2 epidemiological investigations by reliably recovering the prevalence of local circulating variants, even when clinical samples are not available.”

Line 63: Should be "Throughout the COVID-19 pandemic"

Thank you, this has been changed

Line 65: "population health seeking behaviour, test accessibility and testing practices of attending clinicians limit the generalisability of data"

Please provide a citation discussing these potential social biases in health seeking, either for COVID-19 or more generally, beyond asymptomatic cases.

Thank you, the following reference has been cited:

WHO, 2022. Environmental surveillance for SARS-COV-2 to complement public health surveillance – Interim Guidance. <https://www.who.int/publications/i/item/WHO-HEP-ECH-WSH-2022.1> (2022).

Line 121: "using the Sinai protocol with some modifications as described"

What do the authors mean by "as described"? As described in a particular cited source, or described elsewhere in this work? Please specify.

Thank you. We have added: "as described previously", and cited the following reference:

Gonzalez-Reiche, A. S. et al. Introductions and early spread of SARS-CoV-2 in the New York City area. Science (80-.). 369, 297–301 (2020)

Rachida, S. Validating the COVIDSeq and Sinai protocols for wastewater-based sequencing of SARS-CoV-2. In preparation

Line 131: "Again, reads were trimmed and filtered, assembled and mapped"

This sentence has been deleted, as it is redundant.

Line 138 "the generation of consensus sequences from wastewater samples is not meaningful"

Meaningful is a subjective term - the authors should explain what they mean by meaningful here, which is presumably that consensus sequences cannot be interpreted as representing viral haplotypes present in individuals in the population.

Thank you, we have changed this to read as follows:

“the generation of consensus sequences is generally not appropriate because the consensus sequences cannot be interpreted as representing a single viral haplotype present in individuals in population”

Line 142 "R.4.2.0" please specify as R (version 4.2) throughout.

Thank you, this has been added.

Line 172-174 "Signature mutations, which are lineage- defining, nonsynonymous mutations in the spike region of the SARS-CoV-2 genome that are specific to each VOC, (Table S3) were identified in 170 samples (52.3%),"

This does not make sense to me. What mutations were in the 48% of other samples that had sufficient sequencing depth? Lineages of some sort should be identifiable in all samples along with the mutations that correspond to them.

Thank you for this comment. We changed this paragraph as follows:

We identified signature mutations in 170 samples (52.3%), 79 samples from Gauteng, 32 from KwaZulu-Natal, 32 from Free State, 12 from Western Cape, and 15 from the Eastern Cape provinces respectively. The remaining 155 (47.7%) had no signature mutations, and thus could not be independently used to identify lineages in each catchment. We included 143 (44%) samples in our analysis from Gauteng, KwaZulu-Natal, and Free State provinces, and excluded samples from Western Cape, and Eastern Cape due to the small number of samples with signature mutations

Line 175 "Figure 1 illustrates the signature mutations for VOCs that were identified in samples from each wastewater treatment plant"

The version of Figure 1 included in this manuscript is very hard to read and is not fit for publication. It is not included in the PDF, and each panel is included as a separate tif! I do not even know what Figure 1 looks like, or how the panels are arranged. There are panels labeled A-P(!) included as separate TIFs. The panels are very hard to read as the resolution isn't adequate. The legend is only provided in a separate TIF, and there are too many different labels in my opinion. This figure has to be cleaned up in some form for publication.

Thank you for this comment. We agree and have now vastly improved the presentation of our signature mutation analyses from our old Figure 1 (now figure 3). We now provide all data for each province color-coded by variant, and with markers indicating each collection site.

Figure 3 legend reads as follows:

Frequency of signature mutations associated with each variant per epidemiological week, for Gauteng (A), KwaZulu-Natal (B), and Free State (C) provinces. Signature mutations corresponding to each variant are described in Table S3.

We also now provide the mutation level analyses (old Figure 1) as high-resolution, assembled figure in the supplement (Supplementary Figure 2).

Further, the text does not point to specific panels in the figure, so it's a bit unclear where the readers should actually be looking in the figure to find the data supporting the authors' claims in this section.

Thank you. We now specifically describe the data in each component of the figure and compare this with both Freyja-based analyses and clinical surveillance.

“Using these signature mutations, we were able to identify variant waves in samples across our sites (Figure 3A-C, supplementary figure S2). Similar to the Freyja and clinical results, the Beta variant was dominant in samples from April 2021 to June 2021 in all three provinces (Gauteng, KwaZulu-Natal, and Free State). This was followed by the domination of Delta variant and then replaced completely by Omicron BA.1 and thereafter BA.2. In Gauteng Province, Alpha was detected in May, June, and July 2021, while C.1.2 was detected from June to August 2021 (Figure 3A). In KwaZulu-Natal province, Alpha variant mutations were detected in May, and C.1.2 was detected in August 2021 (Figure 3B). In Free State province, we detected one mutation related to Alpha variant in June 2021. Lineage C.1.2 mutations were detected in July, and August 2021 (Figure 3C).”

Section "Detection of SARS-CoV-2 variants from wastewater samples using Freyja tool"

This section is easy to follow. Figure 2 is simple and easy to follow. However, I don't understand why the authors condense their data for this figure. Why not show each sampling location separately, so all can be tracked over time? A version of figure 2 showing changes in lineage frequencies over time for each sampling site separately would communicate the primary results effectively.

Thank you for this comment. We have expanded the analyses in Figure 2 (now Figure 1) to include VOC and lineage level analyses, as well as a direct comparison with clinical surveillance. We have now also provided province level analyses for each of the three sites with the most samples (Gauteng, KwaZulu-Natal and Free State) in the new Figure 2, each of which includes a comparison to clinical surveillance from that province.

I am left wondering about basic questions of the data. Are there any differences in lineages between the sampled locations? Do the authors see introductions of lineages in some sites before others? The authors could expand on their data in many ways here.

Thank you for this comment. We have added the following to the results:

“Across the provinces included in study, we observed some key differences among circulating lineages observed via wastewater. The Alpha variant was detected in Gauteng and Free State in June, but not in KZN. We also observed a small amount of Alpha (about 1% prevalence) in Gauteng in August, but not in KZN or Free State. Delta appeared dominant in the earliest samples taken from KZN, whereas Beta was clearly dominant in Gauteng and Free State. BA.1.1 appeared to play a larger role in Omicron BA.1 spread in Free State than in other provinces, and no BA.2 was observed in Free State during the study. Omicron BA.3 was only detected in Gauteng province, although other BA.1-BA.2 Omicron recombinants were detected both in Gauteng and KZN. We did not detect any BA.2 in Free State, and observed relatively little in KZN, indicative of the BA.2 wave starting earlier in Gauteng. Finally, A.25 and other A lineage circulation was not observed in Gauteng, Free State, or KZN, or in nationwide clinical surveillance, but it was observed in Eastern Cape in June, despite limited wastewater sampling from the province.”

Line 187 " The proportions of lineages were shown in Figure 2b, a total of 68 lineages were detected of which 53 lineages were also reported in clinical cases and 15 lineages were not reported in clinical cases."

Where is the data for this claim? Please cite it here. If an analysis was performed, please include details and results of this analysis.

Which lineages were not reported? How many clinical genomes are the authors pulling? This could be an interesting analysis, but the authors have provided scant details about the results or the methods.

Thank you for this insightful comment. We have added data about the presence of these mutations in clinical sequences from South Africa and globally. The figure has also been improved. Below is what we have changed in the manuscript:

“Of the 411 substitutions/ deletions detected during the study period, 68 were present at >1% prevalence. We used the outbreak.info database to compare those mutations to known published mutations at GISAID of South African sequences during the study period. Out of the 68 mutations, 58 were commonly reported (Table S4). The remaining 10 mutations (S50L, H66Y, T250S, A288T, K444T, Q498H, D627H, L828F, T859N, AND Q1201K), were detected in wastewater despite being present in <1.0% in the sequences of clinical specimens worldwide from GISAID. Further, 7 of the 10 mutations were detected in <1.0% of sequences in GISAID of South African origin (Figure 4B), and three mutations (H66Y, T250S, and T859N) were present in a prevalence ranged between 1-5.5% in clinical sequences from South Africa (Table 1).”

Figure 2: Please present as one figure, not separately. Please show the data for each location separately. What does "Recombinants" mean in Figure 2b?

Thank you. We have combined these into a single figure (now Figure 1) and have now shown this data at the Province level in the new Figure 2 for Gauteng, Kwa-Zulu Natal, and Free State Provinces. We now clarify the meaning of recombinants for both nationwide and the province level analyses. At the national level:

“We also identified cryptic circulation of A lineage viruses including A.25 as well as the Alpha-Delta recombinant lineage XC in June, neither of which had been previously reported in South Africa.”

“We also identified substantial prevalence of BA.3 and other Omicron BA.1-BA.2 recombinant lineages, including XE, XAD, and XAP, that were rarely observed in clinical surveillance during the study period.”

In Gauteng:

“We detected a small population of the XC recombinant in June 2021, but this was not detected in wastewater or clinical surveillance from any other provinces.”

“We also detected the BA.1-BA.2 Omicron recombinant lineages XE, XAD, and XAP.”

In KwaZulu-Natal:

“Omicron BA.2 was detected in wastewater samples in January 2022 for the first time, along with Omicron recombinants XE and XAP, which were also observed in Gauteng wastewater.”

Figure 3: Please show dates on the Y-axis.

We thank the reviewer, this has now been done.

"Figure 4 Dot blot showing the uncommon mutations of SARS-CoV-2 detected in wastewater during304 the period April, 2021 – January, 2022 and their prevalence. The uncommon mutations were detected305 in less than 1% of the sequences from clinical specimens deposited on GISAID during the periods of306 April 2021 to January, 2022."

The reporting of uncommon mutations could be interesting and important, but the authors have provided no details or discussion of them. Were these rare mutations seen in other global lineages, even if they weren't observed locally? Are they consistently seen at some sites in the authors' data more than others? The authors have an opportunity to expand on this data!

Thank you, we have addressed this point above and change the discussion to below:

"Our mutational analysis identified multiple instances of rare mutations in the population (Figure 4B). These mutations were found at a prevalence of >1.0% in wastewater samples but were detected at <0.001% in clinical cases based on the data from GISAID. Although they were uncommon, some mutations were reported previously. Mutation S50L was found to be associated with reduced protein stability³⁵. Mutation Q498H has reportedly caused increased binding affinity of RBD to ACE2³⁶. The presence of uncommon mutations could be explained as "cryptic lineages" as described by Smyth and colleagues³⁷, which are thought to be attributed the cryptic lineages to either un-sampled, possibly chronic, infections or spillover of SARS-CoV-2 from an unidentified animal reservoir."